# Severe Presentation of Congenital Hemolytic Anemias in the Neonatal Age: Diagnostic and Therapeutic Issues

**DOI:** 10.3390/diagnostics11091549

**Published:** 2021-08-26

**Authors:** Valeria Cortesi, Francesca Manzoni, Genny Raffaeli, Giacomo Cavallaro, Bruno Fattizzo, Giacomo Simeone Amelio, Silvia Gulden, Ilaria Amodeo, Juri Alessandro Giannotta, Fabio Mosca, Stefano Ghirardello

**Affiliations:** 1Department of Clinical Sciences and Community Health, Università degli Studi di Milano, 20122 Milan, Italy; valeria.cortesi@unimi.it (V.C.); francescamanzoni.unimi@gmail.com (F.M.); giacomo.amelio@unimi.it (G.S.A.); silvia.gulden@unimi.it (S.G.); fabio.mosca@unimi.it (F.M.); 2Neonatal Intensive Care Unit, Fondazione IRCCS Ca’ Granda Ospedale Maggiore Policlinico, 20122 Milan, Italy; giacomo.cavallaro@policlinico.mi.it (G.C.); ilaria.amodeo@policlinico.mi.it (I.A.); 3UO Ematologia, Fondazione IRCCS Ca’ Granda Ospedale Maggiore Policlinico, 20122 Milan, Italy; bruno.fattizzo@unimi.it (B.F.); jurigiann@gmail.com (J.A.G.); 4Department of Oncology and Hemato-Oncology, University of Milan, 20122 Milan, Italy; 5Neonatal Intensive Care Unit, Fondazione IRCCS Policlinico San Matteo, 27100 Pavia, Italy; s.ghirardello@smatteo.pv.it

**Keywords:** congenital hemolytic anemias, glucose-6-phosphate deficiency, pyruvate kinase deficiency, hereditary spherocytosis, congenital dyserythropoietic anemias, neonatal anemia, jaundice

## Abstract

Congenital hemolytic anemias (CHAs) are a group of diseases characterized by premature destruction of erythrocytes as a consequence of intrinsic red blood cells abnormalities. Suggestive features of CHAs are anemia and hemolysis, with high reticulocyte count, unconjugated hyperbilirubinemia, increased lactate dehydrogenase (LDH), and reduced haptoglobin. The peripheral blood smear can help the differential diagnosis. In this review, we discuss the clinical management of severe CHAs presenting early on in the neonatal period. Appropriate knowledge and a high index of suspicion are crucial for a timely differential diagnosis and management. Here, we provide an overview of the most common conditions, such as glucose-6-phosphate dehydrogenase deficiency, pyruvate kinase deficiency, and hereditary spherocytosis. Although rare, congenital dyserythropoietic anemias are included as they may be suspected in early life, while hemoglobinopathies will not be discussed, as they usually manifest at a later age, when fetal hemoglobin (HbF) is replaced by the adult form (HbA).

## 1. Introduction

Congenital hemolytic anemias (CHAs) are a group of intrinsic red blood cells (RBCs) disorders that share similar clinical and laboratory features related to the reduced RBCs lifespan [1].

CHAs are classified based on the pathophysiology [1,2] (Table 1):
Hemoglobin (Hb) disorders (hemoglobinopathies);Erythrocyte membrane/cytoskeleton defects (membranopathies);RBCs enzyme deficiencies (enzymopathies);Defective erythropoiesis.

In this review, we will address the clinical management of the most frequent CHAs presenting in neonatal age, such as G6PD deficiency, pyruvate kinase deficiency (PKD), and hereditary spherocytosis (HS), with a practical and clinical approach to the differential diagnosis of unconjugated bilirubin-induced jaundice. The discussion of hemoglobinopathies is out of the scope of this review, as, due to the high concentration of fetal hemoglobin in neonatal age, they usually present later in life. Congenital dyserythropoietic anemias (CDAs), although classified as disorders of erythropoiesis, are characterized by the production of fragile RBCs undergoing peripheral hemolysis [3]. Hence, they are included in this overview, as CDA type I and IV may be rare causes of fetal and neonatal anemias.

CHA symptoms usually appear during infancy or early childhood, although mild cases can remain undiagnosed until adulthood [1]. In newborns, significant hyperbilirubinemia, at times severe enough to cause kernicterus, can mask an underlying inherited disorder, which may remain unidentified and classified as “idiopathic jaundice” [4,5]. The USA Kernicterus Registry has reported that CHAs, particularly glucose-6-phosphate dehydrogenase (G6PD) deficiency and hereditary spherocytosis (HS), are among the leading root causes of kernicterus, second only to ABO incompatibility [4,6].

Later in life, the clinical presentation of CHAs results from chronic compensated hemolysis and is mainly related to anemia, which is usually mild to moderate, with fatigue and pallor, associated with jaundice, splenomegaly, and high reticulocyte count [1]. However, the baseline condition can deteriorate due to an aplastic or hemolytic crisis triggered by infections, drugs, or toxins [1]. In addition, pre-existing disorders can exacerbate the symptoms, such as the co-inheritance of Gilbert syndrome, characterized by impaired liver conjugation of bilirubin and is related to extreme jaundice and gallstones formation [7,8,9,10].

A high index of suspicion is required to achieve early diagnosis. Indeed, timely treatment may limit the clinical and socio-economic burden related to these conditions [3,11]. Therefore, we will provide a clinical framework to address CHAs manifesting in the neonatal age in the following paragraphs.

## 2. Approach to the CHAs in Newborns

### 2.1. The Differential Diagnosis of Neonatal Jaundice

Jaundice is the most common neonatal clinical presentation of CHAs, while anemia and splenomegaly usually occur during infancy [4,12,13].

Neonatologists deal with jaundice daily, as up to two-thirds of newborns experience some degree of jaundice and its severe form is the most frequent cause of re-hospitalization in the first week of life [14,15]. In most cases, unconjugated hyperbilirubinemia is a physiological manifestation resulting from the destruction of the fetal RBCs and the immaturity of the hepatic bilirubin conjugation system. In fact, at birth, hematocrit and RBCs volume per body weight are higher than ever in life, RBCs have a shortened lifespan (60–90 days and 35–50 days, respectively, in term and preterm neonates), and the values of uridine 5′-diphosphoglucuronosyltransferase (UGT1A1), the hepatic enzyme committed to bilirubin conjugation, are approximately 1% of adults [14,16].

Physiological jaundice usually develops after 24 h from birth and, in term infants, generally does not persist beyond 10 days of age, with peak bilirubin between 3 and 5 days after birth [16]. The increase of bilirubin is slower than 0.5 mg/dl/h, and usually, it does not exceed 12–13 mg/dl. In premature newborns, jaundice is even more common, being detectable in roughly 80% of them due to the pronounced immaturity of bilirubin conjugation and elimination systems, and the brain is more susceptible to neurological damage [14]. Nevertheless, physiological jaundice does not need any treatment.

Any time neonatal jaundice does not present the features mentioned above, it should require treatment and further investigation to detect the underlying cause. Unfortunately, around 55–66% of extremely severe jaundice is still classified as “idiopathic”, as the etiology remains unidentified [15].

Hemolytic anemia occurs when RBCs production can not keep up with RBCs destruction, and jaundice is the primary clinical manifestation during the first weeks after birth [17]. After this period, the enzymatic system dedicated to bilirubin conjugation and excretion matures, and thus anemia becomes the major issue [17]. At this age, other factors may concur to the risk of developing anemia, such as the sudden decrease in the erythropoietin (EPO) production at birth, due to the switch in the site of erythropoietin production, from the liver to kidney, and high oxygen levels following the beginning of lung respiration [7,18,19].

Hemolysis should be primarily suspected when jaundice occurs early. This occurs when total serum bilirubin exceeds the ninety-fifth percentile hour-specific nomogram value in the first day of life [15,20,21].

Detecting newborns with hemolytic hyperbilirubinemia is relevant as they have an increased risk of bilirubin-induced neurological damage. Therefore, according to the American Academy of Pediatrics (AAP) Guidelines, they have a lower bilirubin threshold for treatment than infants without additional risk factors [22,23]. For instance, it has been reported that up to one-third of all male neonates who experience jaundice seems to be affected by G6PD deficiency, whereas this condition is rare in female newborns with hyperbilirubinemia [13]. Furthermore, neonates with G6PD deficiency have a four-fold risk of presenting hyperbilirubinemia and three-fold the risk of needing phototherapy compared to healthy neonates [24,25]. However, CHAs newborns may seldomly manifest only mild jaundice, while other symptoms may occur later in life [12,16].

Moreover, the onset time of jaundice in G6PD-deficient newborns is delayed compared to the other CHAs, with a bilirubin peak in the 2°–3° day of life, making G6PD deficiency jaundice hardly distinguishable from the physiological one [26,27,28] In contrast, the most severe cases of CHAs may manifest early in fetal life, with intrauterine growth retardation (IUGR) and nonimmune hydrops fetalis syndrome, requiring in utero transfusions, which may lead to premature birth or even death [16,29,30,31]. Several factors, both genetic and environmental, may contribute to the severity of the clinical symptoms. One of the most relevant is Gilbert syndrome, caused by a common polymorphism in the promoter region of the *UGT1A1* gene; its coexistence with CHAs exposes to the risk of developing extreme neonatal jaundice and kernicterus [8].

Therefore, signs of hemolysis should be assessed while evaluating a newborn with unconjugated hyperbilirubinemia. Tests might show the absence of serum haptoglobin (although it can be found even in healthy newborns), increased lactic dehydrogenase (LDH), elevated end-tidal carbon monoxide (ETCO), elevated levels of carboxyhemoglobin, and hemoglobinuria in the absence of RBCs in the urine analysis [17].

Reference range of the main hematologic parameters in term and preterm neonates are reported in Table 2 [32,33,34]. Hour-specific total serum bilirubin (TSB) nomograms are available elsewhere [23].

As carbon monoxide is produced whenever heme turns into bilirubin, the measurement of the ETCO in exhaled breath could be an indirect marker of hemolysis. Neonatal portable tools measuring ETCO with a single nasal cannula are now available, although their use is not routine yet [17]. Other tests, such as elevated reticulocyte count, elevated immature reticulocyte fraction, and elevated nucleated red blood cell count, might not be significant in the first phases of hemolysis [15].

### 2.2. Patient Work-Up

While caring for a newborn with pathological jaundice, the first step is obtaining an accurate family history of jaundice, anemia, early gallstones, splenectomy, or blood transfusions. For instance, in around 65% of patients with HS, the most common CHAs in Caucasians (reported prevalence of 1 in 1000–2000 births), one parent affected by HS can be retrieved, although mildly symptomatic parents may remain unknown [2,8].

The initial laboratory evaluation includes a complete blood count with erythrocyte indices, reticulocyte count, peripheral blood smear, direct antiglobulin test (DAT), and mother and neonatal blood type. If the neonate requires urgent blood transfusions, it should be recommended to collect blood in an EDTA-anticoagulated tube for later morphological analysis [15]. A negative DAT in a neonate with jaundice and signs of hemolysis strongly suggest an intrinsic erythrocyte defect [16].

Blood parameters that can help to make the diagnosis are mean corpuscular hemoglobin concentration (MCHC) and mean corpuscular volume (MCV) ratios higher than 0.36, which are suggestive for HS [8]. In addition, RBCs distribution width (RDW) is also increased in HS [35]. Indeed, an MCHC > 35.5 g/dl and an RDW > 14% have a 100% specificity for pediatric HS, which is the object of the following paragraph [4]. Figure 1 provides a clinical algorithm for the diagnostic work-up.

### 2.3. Peculiarity of Hereditary Spherocytosis Diagnosis

In HS, RBCs lose their standard biconcave shape and take on a spherical shape, becoming vulnerable to splenic trapping and destruction [36]. In 75% of affected patients, an autosomal dominant pattern of inheritance is present and several mutations in the genes encoding for ankyrin-1, α- and β-spectrin, band 3, or protein 4.2 are responsible for most cases of HS [1,4,17].

The presence of spherocytes in a peripheral blood smear is characteristic of HS, although it is less frequent than in adults and is lacking in one in three neonates [8,12]. Moreover, this finding is not pathognomonic since they can be present even in AB0 incompatibility, the most frequent cause of hemolytic anemia in the neonatal period, which is generally associated with a positive DAT, except in the case of low-titer maternal antibodies [4,15]. However, the persistence of spherocytes associated with a negative DAT is suggestive for HS [18]. Even in non-spherocytic CHAs, blood film analysis may aid differential diagnosis by demonstrating elliptocytes, stomatocytes, etc., typical of other rare membranopathies such as hereditary elliptocytosis and hereditary stomatocytosis, as reviewed elsewhere [1]. The diagnosis of HS is confirmed with an osmotic fragility test (OFT) which, however, should be delayed to at least 3 or 4 months of age when cell morphology appears clearer [15,37]. Neonatal erythrocytes are more elastic and have a larger surface-to-volume ratio, resulting in reduced osmotic fragility than adult erythrocytes [15,16]. However, in severe urgent cases, the eosin-5-maleimide (EMA)-binding test is preferred. This is a flow cytometric test that measures the fluorescence of RBCs labeled with EMA, a substance that binds to band 3 and Rh-related proteins in the erythrocytes’ membrane, resulting in decreased fluorescence intensity in HS [8]. Hence, the EMA test is a highly sensitive, cost-effective test and can be considered a first-line screening test for HS in neonates [15,35,37,38].

In reference centers, the electrophoretic quantification of erythrocyte membrane proteins may lead to a definite diagnosis but is usually deferred to childhood and requires at least 4 weeks of transfusion avoidance [37]. In transfused infants, studying parents’ RBCs can provide valuable information [39]. Finally, the ektacytometer laser-assisted optical rotational cell analyzer (LoRRca MaxSis) can assess the RBC deformability in osmotic gradient conditions (Osmoscan analysis) and is a useful diagnostic tool for RBC membrane disorders [40].

### 2.4. Glucose 6 Phosphate Dehydrogenase Deficiency Diagnosis

The AAP recommends measuring G6PD enzymatic activity in jaundiced neonates whose family history or ethnic or geographic origin suggests a high risk of G6PD deficiency or in every infant with inadequate response to phototherapy [23]. In addition, having a pattern of X-linked inheritance should be suspected, particularly in male infants. Still, the deficiency should not be excluded in females due to the lyonization of the X chromosome [26,27].

G6PD deficiency is the most prevalent disorder among CHAs, affecting 4.9% of the worldwide population [11]. Although traditionally regarded by neonatologists as a condition confined to sub-Saharan Africa, Mediterranean Europe, Middle East, and South East Asia, which are the geographical areas where *Plasmodium Falciparum* malaria is or has been endemic, large-scale global migration led to G6PD deficiency emergence in Western countries [11,13,41].

G6PD deficiency causes hemolysis since the enzyme is the only source of reduced nicotinamide adenine dinucleotide phosphate (NADPH) and glutathione in erythrocytes, protecting the membrane from oxidative stress and reactive oxygen species (ROS) [25,42].

It is essential to keep in mind that G6PD levels can falsely result as normal in the case of markedly elevated reticulocyte count [15]. Reticulocytes have five times higher G6PD activity than older erythrocytes; thus, a diagnostic test should be repeated when the reticulocyte count has normalized [13,38]. Moreover, the World Health Organization (WHO) suggests neonatal screening of G6PD deficiency in the regions where the incidence in males is higher than 3–5% by the fluorescent spot test, a qualitative test where NADPH is detected under ultraviolet light [43]. This test is especially accurate for severe G6PD deficiency, as it provides just “normal” or “abnormal” results. However, intermediate G6PD deficiency, as in heterozygous females, might be missed [27,44,45]. The gold standard to diagnose G6PD deficiency is a quantitative spectrophotometric assay that determines G6PD activity per gram of Hb (normal values 7–10 U/g Hb) [45]. This technique allows to detect all grades of deficiency, but it is not universally available. Furthermore, due to the lengthy turn-around time to get a result, newborns might need treatment for jaundice before a diagnosis is made [45]. Although molecular DNA analysis confirms the diagnosis, it is unsuitable for neonatal screening due to the costs, the time needed, and limited availability [44,46].

### 2.5. Pyruvate Kinase Deficiency Diagnosis

The second more common enzymopathy, PKD, should be suspected when G6PD deficiency and membranopathies have been ruled out. It is a rare disorder, with an estimated prevalence ranging from 0.3 to 5 out of 100,000 live births, probably underestimated due to misdiagnosis, subclinical presentation, or early intra-uterine death [9,47]. As with any autosomal recessive disorder, PKD may be more common in some communities, such as the Pennsylvania Amish and the Romani, due to a founder effect or, less frequently, to consanguinity [48,49]. PKD causes hemolysis since RBCs’ metabolism is entirely dependent on glycolysis to produce adenosine triphosphate (ATP) [9], and PK catalyzes the conversion of phosphoenolpyruvate to pyruvate, generating one molecule of ATP [47,50]. The youngest erythrocytes are the most damaged in PKD, as they need a large amount of ATP to survive [51].

PKD diagnosis is based on the measurement of PK activity in RBC lysates by spectrophotometric assay [29]. This is a fast and cheap essay; however, an interval of at least 4 weeks is required in previous RBCs transfusion [9]. False-negatives may occur in the case of incomplete platelets and leucocytes removal and/or marked reticulocytosis, as young erythrocytes contain higher enzyme levels [9,52]. Therefore, the diagnosis should be suspected in normal PK levels that appear low compared with other RBC enzymes [29]. Since PK activity does not predict the severity of the disease, DNA analysis of the PKLR gene is recommended to confirm the diagnosis [9]. Genetic testing requires a small amount of blood, being suitable for newborns or fetuses in prenatal diagnosis, and it can be performed even after blood transfusions [9]. However, the disadvantage of this technique is that around 20% of the mutations founded are not classified as pathogenetic yet, so they are called variants of unknown significance (VUS) [29,53]. In the case of VUS, PK activity by spectrophotometry must be performed. Therefore, the diagnosis should be established by assessing PK activity and the genotyping *PKLR* gene [9].

If the diagnosis of a suspected CHA remains unclear, additional assays (i.e., ektacytometry, rare enzymes assays, and molecular analysis) should be performed under the guidance of an expert hematologist. Genetic analysis can be used as a confirmatory test in complicated cases with no family history [9]. In addition, next-generation sequencing (NGS) panels targeting mutations associated with CHAs have been developed and can be very useful in clinical practice [17,19].

### 2.6. Congenital Dyserythropoietic Anemia Diagnosis

Suppose the most common CHAs have been ruled out. In that case, CDAs should be sought in case of jaundice associated with macro- or normocytic anemia, especially if associated with a suboptimal reticulocyte response, splenomegaly, and/or congenital skeleton malformations, particularly of the hands and feet with syndactyly, polydactyly, and dysplastic nails, in a neonate with a history of IUGR [3,54,55]. CDAs are rare disorders, with a reported incidence of 0.5 cases per million people, probably underdiagnosed due to clinical heterogeneity or misdiagnosis [56]. Traditionally, three major types of CDAs (types I, II, III) have been identified basing on morphological abnormalities of bone marrow erythroblasts [3]. Recently, two new variants that do not fit into the three common types have been described (CDA-IV, XLTDA) and added to the classification [57]. Bone marrow examination is usually performed following standard blood investigation, and it is typically hypercellular, with erythroid hyperplasia and increased erythropoietic/granulopoietic ratio [58,59]. Diagnosis is confirmed through NGS, which allows the analysis of several sequences of DNA in a fast and cheap way [60]. Targeted CDA panels investigate the causative genes of CDAs and could provide earlier diagnosis and even avoid unnecessary invasive examination such as bone marrow aspiration [55]. However, due to pronounced heterogeneity in phenotype expression, a genetic diagnosis cannot provide prognostic information [55].

## 3. Complications

The major complication of severe hyperbilirubinemia is acute bilirubin encephalopathy (ABE) and the subsequent permanent and chronic sequelae are referred to as kernicterus. The overall incidence of kernicterus is estimated at around 0.4–2.7 per 100,000 births in industrialized countries, while it is higher in developing countries, where prompt access to phototherapy and exchange transfusions are hampered by the lack of resources [25]. In the United States, around 20% of cases of kernicterus are linked to G6PD deficiency and, in those patients, mortality and morbidity are increased due to the neurological sequelae [6,24,27,61,62]. The bilirubin-induced damage is mediated by unconjugated bilirubin accumulation in the brain tissue, mainly the basal ganglia. The clinical presentation is characterized by hypotonia alternating with progressive hypertonia of extensor muscles, drowsiness, feeding difficulties, and irritability with a high-pitched cry, which later evolves into lethargy, seizures, fever, and even death. The bilirubin-induced neurologic dysfunction (BIND) score can help in the assessment of the severity of ABE (Table 3) [6]. Therefore, a rapid decrease in bilirubin concentration is mandatory through intensive phototherapy and exchange transfusion to avoid permanent neurological damage. Classical kernicterus is characterized by motor symptoms, with abnormal movements and tone, spasticity, choreoathetosis, auditory neuropathy with or without hearing loss, oculomotor impairments, and dental enamel dysplasia [25]. The milder form of chronic bilirubin encephalopathy may include cognitive dysfunction and learning impairment [6].

Strategies to prevent kernicterus include a careful perinatal history to identify risk factors, universal pre-discharge bilirubin determination, post-discharge follow-up, and parental counseling on hyperbilirubinemia [44].

## 4. Treatment

Current management of CHAs (Table 4) is mainly supportive and aimed at preventing the consequences of severe jaundice and anemia [3].

Phototherapy and exchange transfusion are the hallmarks of neonatal jaundice management, irrespective of the cause. The threshold of bilirubin over which treatment is necessary depends on gestational age and additional risk factors. AAP and National Institute for Health and Care Excellence (NICE) have provided guidelines to help healthcare professionals in jaundice treatment with specific gestational age and post-natal age-graphs [23,63]. Phototherapy and exchange transfusion are needed in 93% and 46% of cases of neonatal jaundice in PK-deficient patients, respectively [9,64]. Differently, a part of G6PD-deficient patients might not experience hyperbilirubinemia in the neonatal period; usually, they develop symptoms later in life or can even be asymptomatic throughout their lifetime [65]. 

### 4.1. Phototherapy

Phototherapy is always the first-line treatment in the case of indirect hyperbilirubinemia. It consists of placing the newborn’s body at a 40–50 cm distance from a light source. The efficacy of phototherapy is influenced by the body surface area exposed, by the wavelength of the light used (460–490 nm is the spectrum suggested), by the irradiance (should be at least 30 µW·cm^−2^·nm^−1^), and by the TSB levels [14,66,67,68]. Phototherapy allows the reduction of TSB through three different mechanisms: 1. Structural isomerization converts bilirubin into lumirubin, a structural isomer that is water-soluble and then excreted with bile and urine without requiring liver conjugation; 2. Photo-oxidation of bilirubin to polar molecules, excreted in the urine 3. Configurational isomerization of native bilirubin isomer to more water-soluble and less toxic isomers [66,67]. Lumirubin is the major mechanism of bilirubin elimination with phototherapy [16]. This therapy is usually safe; rare acute side effects include diarrhea, erythematous rash, dehydration, temperature instability, and skin discoloration. The *bronze baby* syndrome occurs when phototherapy is administered to a cholestatic newborn with direct hyperbilirubinemia [14,67,69]. Long-term adverse effects on growth, development, and behavior have not been reported [70].

### 4.2. Exchange Transfusion

Exchange transfusion is the treatment of choice for severe hyperbilirubinemia according to the guidelines mentioned above. Usually, the values of TSB for which exchange transfusion is necessary to exceed by 4–6 mg/dl are the threshold for phototherapy and are influenced by the hour-specific total bilirubin level, response to intensive phototherapy, serum bilirubin-to-albumin ratio, and the presence of comorbidities such as hemolytic disease, sepsis, acidosis or other acute conditions [16]. Exchange transfusion is always recommended in case of symptoms and signs of acute bilirubin encephalopathy [69,71]. Being an emergency procedure, expert neonatologists in intensive care should perform it. In essence, toxic bilirubin is removed from circulation by taking small aliquots of the baby’s blood, which is replaced with the same amount of donor’s red cells. The procedure needs at least one central line, and the amount of blood exchanged is usually twice the circulating blood volume of the patient. As albumin binds to bilirubin, reducing its toxicity, the albumin/bilirubin ratio should be considered as a severity index. AAP suggests a ratio of 8 in term neonates without other additional risk factors. In selected cases, infusion of albumin should be considered before exchange transfusion to increase the amount of bilirubin removed [69,71]. The main complications are hypoglycemia, infection, portal venous thrombosis, anemia, thrombocytopenia, necrotizing enterocolitis, hypocalcemia and electrolytes disorders, graft versus host disease, and even death, and occur with an overall 12% rate [14,69,71].

### 4.3. Other Measures

Other pharmacological treatments, except for intravenous immune globulin (IVIG) in infants with AB0 or Rhesus immunization, are not recommended [71]. Interestingly, the Sn-mesoporphyrin, a metalloporphyrin that blocks the first enzyme responsible for bilirubin production and acts as a photosensitizer, has been demonstrated to prevent severe hyperbilirubinemia in newborns with G6PD deficiency, thus emerging as a promising research field [16,72].

Since major bilirubin neuronal toxicity is related to oxidative damage, especially in G6PD-deficient patients, antioxidant therapy with vitamin E has been proposed in jaundice neonates, with poor results [25]. Another therapeutic strategy is to restore the GSH/GSSG balance through *n*-acetylcysteine (NAC), a GSH precursor, showing encouraging results in rat models [73,74]; however, it was not confirmed in preterm infants [75].

### 4.4. Transfusion Support

Most neonates affected by CHAs require packed RBCs transfusions, usually in the first year of life [9,19,37]. G6PD-deficient patients do not usually need transfusions in the neonatal period, except for severe hemolysis, resulting from exposure to oxidant products during pregnancy or breastfeeding as in maternal ingestion of fava [16,61,76]. Newborns who received intrauterine blood transfusion might have a delayed nadir of the physiologic decrease of hemoglobin. Therefore, they might not present anemia in the first month of life but later on when the effect of the transfusion is over [29]. In HS patients, transfusion-dependence after 1 year of age is unusual, being present in around 18% of patients, and indicates severe HS [8,18].

### 4.5. Erythropoiesis-Stimulating Agents

Recombinant human erythropoietin (rHuEPO) has been used at the different dosage and posology in HS to evoke bone marrow response during the hypoplastic phase after birth, thus reducing or, less frequently, avoiding the need for RBCs transfusions [8,37]. However, given the cost of the therapy, twice as much as a single transfusion, and the logistic burden—three times weekly injection—further larger and randomized trials are needed to define the best practice [77]. rHuEPO did not lead to the expected outcome on anemia in CDAs [55].

## 5. Conclusions and Future Perspectives

CHAs are a rare cause of anemia in the neonatal age and, due to unspecific and overlapping clinical features, they are frequently misdiagnosed with potential impact on treatment and follow-up. Therefore, the diagnosis still relies on clinical suspicion, careful evaluation of neonatal jaundice, family history, hemolytic markers, and peripherical blood smear morphology. Additionally, laboratory tests including OFT, EMA binding, RBC membrane and enzyme assays, and RBC deformability by ektacytometry contribute to the diagnostic definition of more or less rarer forms.

In recent years, much effort has been carried out to investigate the causative genes and mutations underlying CHAs, allowing molecular diagnosis in an increasing number of patients. However, the genetic study is not advised for common conditions such as HS and G6PD deficiency, where hundreds of mutations have been described with unclear genotype–phenotype correlations. Moreover, in rarer forms such as CDAs, the genetic defect remains unknown in around half of the patients affected [3]. Conversely to traditional diagnostics, genetic testing is feasible even in transfused patients and may replace the need for invasive procedures, such as bone marrow biopsy. The increasing knowledge in this field could also provide prenatal diagnosis and proper counseling of affected fetuses. NGS-based diagnosis showed that approximately 10–40% of initial clinical diagnoses were incorrect and that nearly half of patients with CDAs are affected by RBCs enzymatic defects, mainly PKD [78]. The validation of targeted-panel NGS for hemolytic anemias will possibly become the preferred approach in the near future, although it requires updating panels to include recently discovered causative genes [78]. The use of whole-exome sequencing (WES) and whole-genome sequencing (WGS), which deliver a significant number of VUS whose pathogenicity has to be proven with functional tests, will be likely reserved to investigate undiagnosed cases [78].

The optimization of neonatal diagnosis of CHAs could allow innovative therapeutic strategies aimed at improving long-term outcomes. Emerging therapies specifically directed to correct or modify the defective protein or enzyme have been proposed in G6PD deficiency and PKD [25]. For example, Mitapivat, AG-348, a small molecule PK activator that enhances glycolytic flux, showed promising results in PK-deficient adults, causing increased hemoglobin levels and a concomitant reduction of hemolysis markers. However, data on safety and efficacy in children are scanty [29,79,80]. Another promising therapy for PKD patients is gene therapy via a lentiviral vector, currently under investigation in a phase 1 clinical trial [9,29,81,82].

Lastly, due to the high prevalence of G6PD deficiency, the neonatal screening test is an option, and the WHO suggests its adoption in countries where the prevalence is higher than 3–5% of the male population. Furthermore, as the geographical distribution of G6PD is changed, resulting in a worldwide spreading, the early screening might be beneficial in other countries to prevent bilirubin-induced neurological damage [27]. Although, of course, the cost-effectiveness of such a test has to be established, it holds promise for future applications.

## Figures and Tables

**Figure 1 diagnostics-11-01549-f001:**
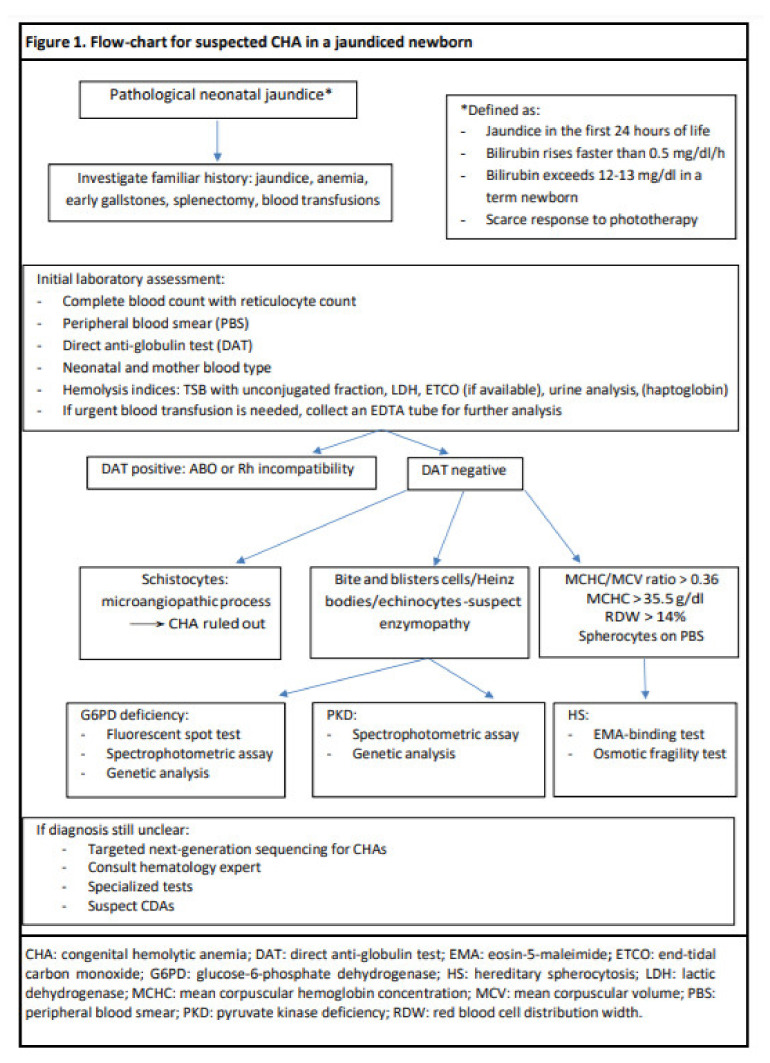
Flow-chart for suspected CHA in a jaundiced newborn.

**Table 1 diagnostics-11-01549-t001:** Classification of the most common congenital anemias based on pathophysiology [1].

Membranopathies	Enzymopathies	Hemoglobinopathies	Defective Erythropoiesis
Hereditary spherocytosis Hereditary elliptocytosis Hereditary pyropoikilocytosis Hereditary stomatocytosis	Glucose 6 phosphate dehydrogenase deficiency Pyruvate kinase deficiency Glucose phosphate isomerase deficiency Hexokinase deficiency	Sickle cell disease Alpha thalassemia Beta thalassemia Unstable hemoglobin	Congenital dyserythropoietic anemias type I Congenital dyserythropoietic anemias type II Congenital dyserythropoietic anemias type III Congenital dyserythropoietic anemias type IV

**Table 2 diagnostics-11-01549-t002:** Reference ranges of the main hematologic parameters in term and preterm neonates.

	Preterm			Term			References
	Day 1	Day 7	Day 28	Day 1	Day 7	Day 28	
RBC (× 10^6^ per µL)	4.71 ± 0.75	4.45 ± 0.83	3.17 ± 0.6	5.14 ± 0.7	4.86 ± 0.6	4.00 ± 0.6	
Hb (g/dl)	18.2 ± 2.7	16.3 ± 2.9	10.9 ± 1.9	19.3 ± 2.2	17.9 ± 2.5	14.2 ± 2.1	
Hct (%)				61 ± 7.4	56 ± 9.4	43 ± 5.4	
MCV (fl)	115 ± 5	110 ± 5	100 ± 5	119 ± 9.4	118 ± 11.2	105 ± 7.5	
MCH (pg)	38.9 ± 1.7	37.3 ± 1.8	35.1 ± 1.9				
MCHC (g/L)	33.5 ± 1.2	33.9 ± 1.3	34.4 ± 1.0	31.6 ± 1.9	32 ± 1.6	33.5 ± 1.6	
Reticulocyte (%)				3.2 ± 1.4	0.5 ± 0.4	0.6 ± 0.3	
Haptoglobin (mg/dl)	5.9 (cord blood)	3.43		4.62 (cord blood)			Chavez-Bueno et al., 2011 [33]
LDH (U/L)				380–818			Lackmann et al. 1993 [34]

Adapted from Neonatal Hematology, de Alarcòn P A and Werner E J, 2005 [32]. Hb, hemoglobin; Hct, hematocrit; LDH, lactate dehydrogenase; MCH, mean corpuscular hemoglobin; MCHC, mean corpuscular hemoglobin concentration; MCV, mean corpuscular volume; RBC, red blood cells. Data expressed as mean ± SD, except for haptoglobin that is median and LDH that is fifth–ninety-fifth percentile.

**Table 3 diagnostics-11-01549-t003:** Bilirubin-Induced Neurologic Dysfunction (BIND) score [6].

Mental status	Normal	0
	Sleepy but arousable, decreased feeding	1
	Lethargic, poor suck, irritable, and/or jittery	2
	Semi-coma, apnea, unable to feed, seizure, coma	3
Muscle tone	Normal	0
	Persistent mild to moderate hypotonia	1
	Mild to moderate hypotonia alternating with hypertonia	2
	Persistent retrocollis and opisthotonos	3
Cry pattern	Normal	0
	High-pitched when aroused	1
	Shrill, hardly consolable	2
	Inconsolable crying or weak/absent cry	3

Score of 1–3: mild ABE. Score of 4–6: moderate ABE. Score of 7–9: advanced ABE. ABE: acute bilirubin encephalopathy.

**Table 4 diagnostics-11-01549-t004:** Summary of the CHAs reviewed in this article: neonatal and pediatric features.

	Etiology and Inheritance Pattern	Clinical Manifestations	Diagnosis	Treatment
Glucose-6-phosphate dehydrogenase (G6PD) deficiency	*G6PD* gene (Xq28) X-linked	Neonatal jaundice Acute hemolytic anemia	Fluorescent spot test (qualitative) Spectrophotometric assay (quantitative, gold standard); genetic test	Phototherapy and exchange transfusion (neonatal jaundice); RBCs transfusion and supportive care in AHA
Pyruvate kinase deficiency (PKD)	*PKLR* gene (1q21) Autosomal recessive	IUGR, hydrops fetalis, neonatal jaundice Chronic hemolytic anemia; splenomegaly	Spectrophotometric assay Genetic test (both needed)	Phototherapy and exchange transfusion (neonatal jaundice); RBCs transfusion splenectomy; folic acid supplementation
Hereditary spherocytosis (HS)	ANK, SLC4A1, SPTB, SPTA1, EPB42 genes 75% autosomal dominant	Neonatal jaundice and anemia; chronic hemolytic anemia; splenomegaly	Clinical signs + laboratory findings (high MCHC, spherocytes) EMA binding test OFT Genetic test	Phototherapy and exchange transfusion (neonatal jaundice); RBCs transfusion; rhEPO; splenectomy; folic acid supplementation
Congenital dyserythropoietic anemias (CDAs)	CDA-Ia: *CDAN1* gene (15q15.2) aAutosomal recessive	Hydrops fetalis, IUGR, neonatal jaundice Moderate to severe macrocytic anemia Hepatosplenomegaly Skeleton involvement	Morphological analysis at bone marrow examination; genetic test (targeted CDAs panels) Ham test positive in CDA-II Hypoglycosylation of band 3 or 4.5 membrane proteins in RBCs in CDA-II; increased HbF in CDA-IV targeted NGS panels	RBCs transfusion; folic acid and vitamin B12 INF-a in CDA-I Splenectomy in CDA-II Stem cell transplant
	CDA-Ib: *C15orf41* gene (15q14) Autosomal recessive			
	CDA-II: *SEC23B* gene (20p11.23) Autosomal recessive	Mild to severe normocytic anemia		
	CDA-III: *KIF23* gene (15q21) Autosomal dominant	Mild anemia		
	CDA-IV: *KLF1* gene (19p13.2) Autosomal dominant	Bleeding manifestations Severe anemia		
	XLTDA: *GATA1* gene (Xp11.23) X-linked	Mild to severe anemia		

EMA: eosin-5-maleimide; HbF: fetal hemoglobin; INF-a: alpha-interferon; IUGR: intrauterine growth restriction; MCHC: mean corpuscular hemoglobin concentration; NGS: next-generation sequencing; OFT: osmotic fragility test; RBC: red blood cells; rhEPO: recombinant human erythropoietin.

## Data Availability

Not applicable.

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
