# Peer review of "Severe Presentation of Congenital Hemolytic Anemias in the Neonatal Age: Diagnostic and Therapeutic Issues"

_diagnostics, 2021, doi:10.3390/diagnostics11091549_

Round 1

Reviewer 1 Report

very well written and interesting review.

i have only one major comment: it would be very useful for readers if the authors would add a table with normal values (in preterm and term neonates) for Hb levels, reticulocyte counts, LDH etc etc

Author Response

Thank you for your comment and time for revision. We have added table 2 reporting reference ranges as suggested (see page 3, lines 172-175 and page 4 line 191). We have added additional references supporting the data reported.

Reviewer 2 Report

The authors present a very informative review of congenital haemolytic anaemia in newborn infants. The manuscript is helpful for individuals new to the subject as well as an excellent point of reference for experienced paediatric haematologists and laboratory staff wanting to learn about the clinical features of these diseases.

Minor point. The sequence of references in the text is 1, 2, 3 then goes to 6, 7,8.  Reference 5 appears before 4. 

Author Response

Thank you for your comment and your time for revision. We have addressed the minor point raised.